# Generating Safety-Critical Automotive C-programs using LLMs with Formal Verification

**Merlijn Sevenhuijsen**[1,2]                                MERLIJN.SEVENHUIJSEN@SCANIA.COM
**Gustav Ung**[1,2]                                          GUSTAV.UNG@SCANIA.COM
**Minal Suresh Patil**[1]                                    MINAL.PATIL@SCANIA.COM
**Mattias Nyberg**[1,2]                                      MATTIAS.NYBERG@SCANIA.COM

[1] *Scania CV AB, Södertälje, Sweden*

[2] *KTH Royal Institute of Technology, Stockholm, Sweden*

**Editors:** Leilani H. Gilpin, Eleonora Giunchiglia, Pascal Hitzler, and Emile van Krieken

## Abstract

We evaluate the feasibility of generating formally verified C code that adheres to both functional and non-functional requirements using Large Language Models (LLMs) for three real industrial, automotive safety-critical software modules. We explore the capabilities of ten LLMs and four prompting techniques — Zero-Shot, Zero-Shot Chain-of-Thought, One-Shot, and One-Shot Chain-of-Thought — to generate C programs for the three modules. Functional correctness of generated programs is assessed through functional verification, and adherence to non-functional requirements is evaluated using an industrial static analyzer, along with human evaluation. The results demonstrate that it is feasible for LLMs to generate functionally correct code, with success rates of **540/800**, **59/800**, and **46/800** for the three modules. Additionally, the generated programs frequently adhere to the defined non-functional requirements. In the cases where the LLM-generated programs did not adhere to the non-functional requirements, deviations typically involve violations of single-read and single-write access patterns or minimal variable scope constraints. These findings highlight the promise and limitations of using LLMs to generate industrial safety-critical C programs, providing insight into improving automated LLM-based program generation in the automotive safety-critical domain.

## 1. Introduction

Large Language Models (LLMs) have demonstrated astonishing capabilities in generating code from natural language specifications (Vaithilingam et al., 2022; Ross et al., 2023). Furthermore, LLMs have proven to be valuable in software development tasks, from code generation and debugging to documentation creation (Wu et al., 2023; Huang et al., 2023, 2024). However, several studies discuss limitations related to guaranteeing the quality and correctness of generated code (Tambon et al., 2024; Zhong and Wang, 2024; Lin et al., 2025). This limits the applicability of LLMs in domains where correctness is vital, such as in the safety-critical automotive software domain.

Recognizing challenges in guaranteeing quality and correctness, this paper examines whether LLMs can generate functionally correct *C programs* that adhere to both *functional* and *non-functional* requirements for *industrial automotive embedded* software. To guarantee adherence to functional requirements, we formally verify that the generated programs meet formal specifications written in ANSI/ISO C Specification Language (ACSL) (Baudin et al., 2021), a widely adopted annotation language for C. Regarding Non-Functional Requirements (NFRs), it is important to note that industrial safety-critical code must adhere to a large set of NFRs, as demanded by safety standards, such as ISO 26262 (ISO, 2018). For example, in automotive software, compliance with standardized coding guidelines, such as MISRA-C (MIRA Ltd, 2004), is a common requirement. Furthermore, the architectural framework used in a particular software development context typically requires following specific syntax and interface requirements. To evaluate adherence to NFRs,

we combine the use of an industrial static code analysis tool with manual inspection. To avoid biased results caused by validation examples being part of the training data of the LLMs, we run all experiments on code examples that have not been previously published.

Several previous papers have explored the use of formal verification methods to ensure that LLM-generated code meets both *functional* and *non-functional requirements*. To ensure functional correctness, (Patil et al., 2024), (Mukherjee and Delaware, 2024), and (Slama and Lemire, 2024) integrate LLMs with formal verification tools to assess the functional correctness, safety, and potential vulnerabilities of LLM-generated code. The work (Cramer and McIntyre, 2025) explores the feasibility of using formal software verification, specifically the SPARK framework for Ada, to ensure the reliability of LLM-generated code. Sevenhuijsen et al. (2024) introduce a tool that combines LLMs with formal verification to automatically and iteratively generate formally verified C programs. The tool is evaluated on problems presented in Codeforces competitions. Regarding adherence to NFRs, (Almonte et al., 2025) find that LLMs can generate NFR specifications aligning with industry-standard models. However, Lin et al. (2025) demonstrate that prompting LLMs to meet NFRs may also negatively impact functional correctness, reflecting inherent trade-offs.

Although previous studies have begun investigating the use of LLMs for safety-critical software, a comprehensive exploration to ensure both functional correctness and adherence to NFRs for LLM-generated industrial code remains limited. Specifically, previous studies have not simultaneously assessed adherence to both functional and non-functional requirements for the safety-critical automotive domain. In our earlier work (Patil et al., 2024), we investigated the generation of C programs for three real industrial safety-critical modules. The present paper provides three main extensions: (1) exploring additional prompting techniques such as One-Shot; (2) broadening the evaluation by evaluating ten recent LLMs; and (3) significantly extending the assessment of adherence to NFRs, like interface consistency and adherence to safety-critical coding requirements.

The paper is organized as follows. Section 2 describes the case studies that form the basis of the experiments. Section 3 presents the experimental design criteria, covering the creation of specification types, the selection of LLMs, the design of prompts, and methods for evaluating generated programs. Research questions are defined in Section 4. Section 5 presents and discusses the experimental results, followed by a discussion in Section 6. The paper is concluded in Section 7.

## 2. Industrial Case Studies

To evaluate the capabilities of LLMs in the context of the safety-critical automotive domain, we utilize three *application software modules* from real-world automotive C code developed by the heavy-vehicle manufacturer Scania. These application software modules perform essential functions in embedded systems, where correctness is critical for ensuring safe and reliable vehicle operation. Throughout the paper, we refer to these application software modules as *modules* for conciseness.

Modules execute within an electronic control unit (ECU), where a static scheduler invokes the main function of the module at fixed intervals. The execution is strictly sequential, ensuring deterministic behavior by limiting recursion and dynamic memory allocation.

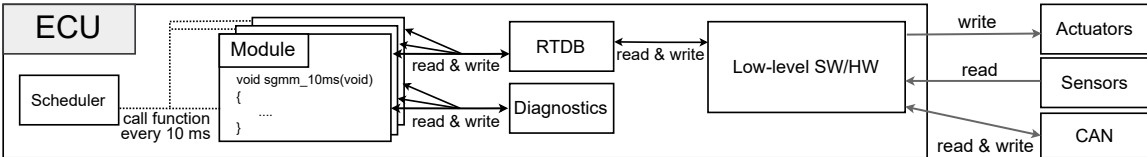

Figure 1: ECU architecture with internal and external interactions.

A module interacts with multiple components in an automotive embedded system, as shown in Figure 1. The Real-Time Database (RTDB) manages communication between different modules, allowing a module to read and write shared signals. The diagnostics module handles fault detection

and logging. The low-level software (SW) and hardware (HW) (Low-Level SW/HW) abstraction layer provides access to peripherals, sensors, and actuators. External communication with other ECUs occurs via the Controller Area Network (CAN) (ISO, 2015). Each module reads inputs from the RTDB, representing sensor values, control signals, or stored internal states. The computed outputs are written back to the RTDB or sent to the diagnostics module. We select three modules with natural language specifications from Scania's development based on four criteria: (1) the module covers diverse functionalities; (2) the module includes a manually written and formally verified *reference implementation* ranging from 50 to 150 Lines of Code (LoC), balancing feasibility for LLM generation with sufficient complexity for evaluation; (3) the functional requirements are formally verifiable using Frama-C, and (4) the module performs safety-critical functions, requiring high reliability and correctness.

To enable verification, we structure each module as a single translation unit, abstracting interactions with other components such as a RTDB. These abstractions are made by replacing functionality with globally defined static variables. Crucially, the safety-critical nature and functionality of the code remain consistent with the original implementation taken from Scania's repository.

| Module | #ACSL Clauses | #NL Requirements | #LoC Reference |
|--------|---------------|------------------|----------------|
| SGMM | 13 | 5 | 89 |
| BRAK | 7 | 5 | 108 |
| SFLD | 59 | 10 | 102 |

Table 1: Overview of the safety-critical modules used as case studies.

Table 1 provides an overview of the three chosen modules. Column "#ACSL Clauses" details the number of ACSL clauses present in the formal specification, column "#NL Requirements" presents the number of natural language requirements, and "#LoC Reference" shows the length of the reference implementation in terms of LoC. The *gearbox activation* module (SGMM) ensures the safe engagement of a gearbox component by processing real-time sensor data related to motor speed, magnetic field integrity, and gear position. The *brake light* module (BRAK) determines the activation of brake lights based on system state, remote requests, and voltage levels, handling conflicting signals to ensure reliable operation. The *oil level warning* module (SFLD) monitors oil levels over time and triggers diagnostic warnings when the oil level remains low for a predefined duration, preventing false alarms while ensuring timely fault detection. SGMM and BRAK require fewer ACSL clauses than SFLD, indicating that their verification involves fewer constraints and more straightforward logical conditions.

## 3. Experimental Setup

### 3.1. Prompt Design and LLMs

In the experiments, we invoke LLMs using prompts to generate a program for the modules defined in Section 2. Each generated program is referred to as an *implementation*. The aim is to produce *correct implementations* that satisfy both functional and non-functional requirements. For each module, we define five inputs used in the prompts: (1) ACSL specifications that formally define the expected behavior; (2) natural language specifications describing the intended functionality in English; (3) a function signature that specifies the function name, return type, and parameters; (4) a header file containing the type definitions, struct definitions, and constants used in the *reference implementation*; and (5) an interface definition that specifies RTDB signals and local variables that the implementation can use. Appendix A exemplifies the inputs of a module.

These inputs are then used to construct structured prompts, which provide LLMs with information for generating *implementations* for the specified modules. We define four prompting techniques to investigate how different prompt formats affect the ability of LLMs to generate implementations

that adhere to both functional and non-functional requirements: (1) the Zero-Shot Prompt (Wei et al., 2023), only providing high-level task instructions; (2) the Zero-Shot Chain-of-Thought (CoT) Prompt (Wei et al., 2023), augmenting the instructions with a step-by-step reasoning path; (3) the One-Shot Prompt (Brown et al., 2020), including a single example of a completed module along with its inputs (detailed in Appendix A) to employ in-context learning; and (4) One-Shot-CoT, combining both One-Shot and CoT. An outline of the prompts is provided in Appendix B. Each prompt begins with a *system prompt* that defines how the LLM should behave, followed by a *user prompt* that provides the input to the LLM.

LLMs can leverage *reasoning through prompting*, typically via CoT prompts. In these prompts, the reasoning steps can improve reasoning even in LLMs not specifically optimized for reasoning, such as `GPT-4.1` (OpenAI, 2025a) (*non-reasoning LLM*). In contrast, LLMs exist that are explicitly trained or fine-tuned for *multi-step reasoning*. These LLMs, such as `DeepSeek-R1` (DeepSeek-AI, 2025), have a conversation with itself on how to answer the prompt. We refer to these LLMs as *reasoning LLMs*. Note that for reasoning LLMs, the LLM still reasons, even if the prompt is not a CoT variant. The reasoning steps present in the CoT act as a guide for the reasoning models, but can also be helpful for non-reasoning LLMs.

We evaluate a diverse set of ten LLMs based on their strong performance in recent code generation benchmarks such as RepairBench (Silva and Monperrus, 2024), BigCodeBench (Zhuo et al., 2024), and HumanEval (Chen et al., 2021), as well as their specialization in programming tasks. From experiments conducted with 22 LLMs (listed in Appendix C), we select ten representative LLMs—five closed-source and five open-source— of varying size and providers for focused analysis in this paper. The selection emphasizes the best-performing LLMs, regardless of their cost or whether they are a reasoning LLM or not. Rather than comparing LLMs directly, the focus is on assessing the feasibility of generating at least one implementation adhering to the imposed requirements for each module with recent LLMs.

### 3.2. Evaluation of Generated Implementations

After an LLM generates an implementation, it is compiled. If the compilation is successful, the implementation is evaluated for functional correctness. If successful, it is also assessed with respect to non-functional requirements NFRs. To verify functional correctness, we use the so-called *weakest precondition* (WP) (Baudin et al., 2024) plugin in the state-of-the-art C source code analysis framework Frama-c (Correnson et al., 2025) version 29.0. Frama-c is open-source and employs plugins to formally verify that C programs adhere to formal ACSL (Baudin et al., 2021) specifications. ACSL specifications are placed within special comments (" `/*@ ...*/`") and primarily include function contracts and code annotations.

To assess the adherence to defined NFRs, we use an industrial static analysis tool designed to enforce coding standards and detect deviations from predefined NFRs. The employed static analysis tool supports custom rules and is utilized in the automotive safety-critical domain. Additionally, manual inspection is necessary to verify NFRs that the tool does not cover.

Previous work (Chen et al., 2021; Austin et al., 2021; Yu et al., 2024) assesses functional correctness by executing test cases and computing the unbiased Pass@$k$ metric. In contrast, this study introduces the *Verify@k* metric, which evaluates the feasibility of generating formally verified implementations. Verify@$k$ first checks whether the generated implementation compiles successfully. If compilation succeeds, the implementation undergoes formal verification. An implementation is considered verified under Verify@$k$ only if it passes both compilation and verification. To compute Verify@$k$, we generate $n \geq k$ implementations per module and determine how many of these successfully verify. We use $k \in \{1, 5, 10\}$ to analyze different likelihoods of success. $k = 1$ reflects the effectiveness when generating a single function, while $k = 5$ and $k = 10$ provide insight into whether

multiple attempts are necessary for generating success. From the generated implementations, we count the number of formally verified implementations $c \leq n$ that pass both compilation and verification, using this to calculate the Verify@k metric: $\text{Verify@}k := \underset{\text{Specifications}}{\mathbb{E}} \left[ (1 - \binom{n-c}{k}) / \binom{n}{k} \right]$.

## 3.3. Non Functional Requirements

In general, NFRs refer to safety-critical properties such as performance, security, and maintainability, which do not specify the functional behavior of the code but rather its qualitative attributes. We divide the NFRs into *Code Guidelines* and *Implementation Constraints*. The former specifies requirements such as structured control flow derived from The Rules of Ten (Holzmann, 2006). The latter, derived from experimental observations, captures constraints that ensure compatibility with formal verification and adherence to safety-critical practices. When applicable, we map these NFRs into a subset of MISRA-C (MIRA Ltd, 2004), which static analysis tools can automatically check. All 11 NFRs are summarized in Figure 2 and fully explained in Appendix D.

---

*Non-Functional Requirements 1*

Safety-Critical Code Guidelines (Derived from Rules of 10)
NFR1 (control flow). Restrict all code to simple control flow constructs, without recursion, goto, or jumps.
NFR2 (loops). All loops must have a fixed upper bound.
NFR3 (memory). Do not use dynamic memory allocation after initialization.
NFR4 (scope). Data objects must be declared at the smallest possible level of scope.
NFR5 (validation). Check return values and validate function parameters.
NFR6 (pointers). The use of pointers must be restricted.
Implementation Constraints (Derived from Observations)
NFR7 (RTDB). Read from and write to each RTDB variable at most once.
NFR8 (constructs). Only use standard C constructs or header−defined constructs.
NFR9 (interface). Adhere to the provided interface definition.
NFR10 (imports). Do not use external imports.
NFR11 (formal specification logic). Do not use formal specification logic such as ghost variables or predicates in the implementation.

---

Figure 2: Defined non-functional requirements analyzed in this paper.

## 4. Research Questions and Experimental Methodology

**RQ1 (functional correctness feasibility)**: *In the domain of safety-critical software, how feasible is it to generate a functionally correct implementation using state-of-the-art LLMs, from given functional requirements?* Understanding feasibility is crucial, as it determines whether LLMs can produce functionally correct implementations for use in safety-critical domains, such as automotive software. To assess this, we generate 20 program implementations per prompting technique, LLM, and module, using a temperature of 0.7. This temperature is chosen because prior work has shown it balances diversity and coherence in generated code (Wen et al., 2024; Granberry et al., 2024). Lower temperatures (e.g., 0.1) produce more deterministic output that limits exploration, while higher values (e.g., 0.9) can introduce syntactic or logical errors. For LLMs that do not allow temperature control (i.e. `o4-mini-high`), the temperature is fixed. Each of the generated implementations is compiled, and any that fail this are discarded. The remaining implementations undergo formal verification to verify adherence to the functional requirements outlined in the formal specification. We define that an LLM *solves* the problem of implementing a given module if *at least one* of its generated implementations formally verifies successfully . We then check whether all modules are solved; if so, then we say it is *feasible* to generate functionally correct implementations using LLMs.

**RQ2 (prompting techniques influence)**: *How do different prompting techniques influence the likelihood of generating formally verified programs?* Prompting techniques impact how LLMs generate implementations from requirements. Evaluating the influence of different prompting techniques as defined in Section 3.1 helps determine whether structured instructions improve the success of

generating correct implementations. We employ the Verify@k metric to measure the likelihood of generating at least one formally verified implementation within $k$ attempts, which is calculated for multiple values of $k$ —Verify@1, Verify@5, and Verify@10— across different prompting techniques.

**RQ3 (non-functional compliance feasibility)**: *In the domain of safety-critical software, how feasible is it to generate an implementation adhering to non-functional requirements using state-of-the-art LLMs?* Even if a generated implementation passes formal verification, it may still violate NFRs such as safety-critical coding guidelines. Understanding these deviations is crucial in determining whether LLM-generated implementations are to be used in production. For RQ3, it is deemed feasible only if each module has at least one implementation that is both formally verified and fully compliant with NFRs defined in Figure 2.

Starting with the verified implementations from RQ1, we assess compliance using an industrial static analysis tool, which checks NFR1 (control flow), NFR3 (memory usage), NFR5 (validation), and NFR6 (pointers). To meet NFR8 (interface) and NFR10 (imports), only function bodies (thus all custom-defined constructs and external imports are removed from the implementation) are evaluated. Compliance with NFR11 (formal specification logic) is ensured through compilation and verification, which fail if specification-specific constructs are used.

Since the employed static analysis tool does not cover NFR2 (loops), NFR4 (scope), NFR7 (RTDB), and NFR9 (interface), we must manually inspect implementations to confirm adherence to these NFRs. An exhaustive review of all implementations is infeasible due to time constraints (possibly involving 2,400 implementations), so we allocate a 25-hour budget for manual inspection. Within this time frame, we select a representative subset from every LLM and module, prioritizing cases with fewer successful solves to ensure these underrepresented combinations are also analyzed. Although the selection includes some randomness, finding at least one compliant implementation per module is sufficient to demonstrate that such solutions are possible.

## 5. Experimental Results

**RQ1 (functional correctness feasibility)**: Table 2 summarizes the formal verification outcomes for the ten chosen LLMs across four prompting techniques and three safety-critical modules: SGMM, BRAK, and SFLD. It is structured as follows. The first column, "*Large Language Model*", specifies the name, version, and citation of the employed LLM. Additionally, the LLMs are categorized by source availability (Closed-Source or Open-Source) and whether they are Reasoning or Non-Reasoning LLMs. The second column, "*Prompt Technique*", specifies the prompting technique used as defined in Section 3.1. The remaining columns present the verification success rates for the SGMM, BRAK, and SFLD modules. Each module is associated with four columns presenting four metrics. The first three metrics, "v@1", "v@5", and "v@10", show the verify@k metric as defined in Section 3.2. The fourth metric, "$v_{s/20}$", shows the total number of verified implementations out of 20 generated implementations. For these four metrics, the "-" symbol indicates that none of the 20 generated implementations pass verification. For each module, a total of 800 implementations are generated (20 implementations using four prompting techniques and ten LLMs). The bottom row presents the total number of formally verified implementations for each module. The results in Table 2 indicate that generating formally verified implementations is feasible for all three defined modules. However, the feasibility differs depending on the module. For SGMM, all LLMs are capable of generating at least one formally verified implementation. In total, 540 of the 800 generated implementations are formally verified for SGMM, demonstrating that LLMs can consistently generate this module. The high success rate of SGMM can most likely be attributed to the relative simplicity of generating implementations for the SGMM module. The module does not involve complex logic, and its formal specifications closely align with an implementation.

| Large Language Model | Prompt Technique | SGMM | | | | BRAK | | | | SFLD | | | |
|---|---|---|---|---|---|---|---|---|---|---|---|---|---|
| | | v@1 | v@5 | v@10 | $v_{s/20}$ | v@1 | v@5 | v@10 | $v_{s/20}$ | v@1 | v@5 | v@10 | $v_{s/20}$ |
| **GPT-4.1** | Zero-Shot | 0.75 | 1.00 | 1.00 | 15 | 0.10 | 0.45 | 0.76 | 2 | 0.05 | 0.25 | 0.50 | 1 |
| 2025-04-14 (OpenAI, 2025a) | Zero-Shot-CoT | 0.85 | 1.00 | 1.00 | 17 | 0.05 | 0.25 | 0.50 | 1 | 0.15 | 0.60 | 0.89 | 3 |
| Closed-Source | One-Shot | 0.45 | 0.97 | 1.00 | 9 | 0.05 | 0.25 | 0.50 | 1 | 0.05 | 0.25 | 0.50 | 1 |
| Non-Reasoning | One-Shot-CoT | 0.90 | 1.00 | 1.00 | 18 | 0.30 | 0.87 | 0.99 | 6 | 0.25 | 0.81 | 0.98 | 5 |
| **o4-mini-high** | Zero-Shot | 1.00 | 1.00 | 1.00 | 20 | - | - | - | - | 0.10 | 0.45 | 0.76 | 2 |
| 2025-04-16 (OpenAI, 2025b) | Zero-Shot-CoT | 0.95 | 1.00 | 1.00 | 19 | 0.05 | 0.25 | 0.50 | 1 | 0.05 | 0.25 | 0.50 | 1 |
| Closed-Source | One-Shot | 1.00 | 1.00 | 1.00 | 20 | 0.05 | 0.25 | 0.50 | 1 | - | - | - | - |
| Reasoning | One-Shot-CoT | 0.90 | 1.00 | 1.00 | 18 | 0.05 | 0.25 | 0.50 | 1 | 0.20 | 0.72 | 0.96 | 4 |
| **Grok-3** | Zero-Shot | 1.00 | 1.00 | 1.00 | 20 | 0.20 | 0.72 | 0.96 | 4 | 0.40 | 0.95 | 1.00 | 8 |
| beta (xAI, 2025) | Zero-Shot-CoT | 1.00 | 1.00 | 1.00 | 20 | 0.40 | 0.95 | 1.00 | 8 | 0.25 | 0.81 | 0.98 | 5 |
| Closed-Source | One-Shot | 0.05 | 0.25 | 0.50 | 1 | - | - | - | - | - | - | - | - |
| Reasoning | One-Shot-CoT | 0.95 | 1.00 | 1.00 | 19 | 0.50 | 0.98 | 1.00 | 10 | 0.15 | 0.60 | 0.89 | 3 |
| **Gemini-2.5-Pro** | Zero-Shot | 0.85 | 1.00 | 1.00 | 17 | 0.10 | 0.45 | 0.76 | 2 | 0.05 | 0.25 | 0.50 | 1 |
| 05-06 (DeepMind, 2025b) | Zero-Shot-CoT | 1.00 | 1.00 | 1.00 | 20 | - | - | - | - | 0.15 | 0.60 | 0.89 | 3 |
| Closed-Source | One-Shot | 0.85 | 1.00 | 1.00 | 17 | - | - | - | - | 0.20 | 0.72 | 0.96 | 4 |
| Reasoning | One-Shot-CoT | 0.95 | 1.00 | 1.00 | 19 | 0.05 | 0.25 | 0.50 | 1 | 0.15 | 0.60 | 0.89 | 3 |
| **Sonnet-3.7** | Zero-Shot | 1.00 | 1.00 | 1.00 | 20 | 0.10 | 0.45 | 0.76 | 2 | - | - | - | - |
| 20250219 (Anthropic, 2025) | Zero-Shot-CoT | 1.00 | 1.00 | 1.00 | 20 | 0.35 | 0.92 | 1.00 | 7 | 0.10 | 0.45 | 0.76 | 2 |
| Closed-Source | One-Shot | 0.15 | 0.60 | 0.89 | 3 | - | - | - | - | - | - | - | - |
| Reasoning | One-Shot-CoT | 1.00 | 1.00 | 1.00 | 20 | 0.10 | 0.45 | 0.76 | 2 | - | - | - | - |
| **Qwen2.5-Coder-32B-Instruct** | Zero-Shot | 0.95 | 1.00 | 1.00 | 19 | - | - | - | - | - | - | - | - |
| 2024-11-12 (Yang et al., 2024) | Zero-Shot-CoT | 0.50 | 0.98 | 1.00 | 10 | - | - | - | - | - | - | - | - |
| Open-Source | One-Shot | 0.40 | 0.95 | 1.00 | 8 | - | - | - | - | - | - | - | - |
| Reasoning | One-Shot-CoT | 0.45 | 0.97 | 1.00 | 9 | - | - | - | - | - | - | - | - |
| **DeepSeek-R1-Distill-Qwen-32B** | Zero-Shot | 0.70 | 1.00 | 1.00 | 14 | 0.05 | 0.25 | 0.50 | 1 | - | - | - | - |
| 0121 (DeepSeek-AI, 2025) | Zero-Shot-CoT | 0.30 | 0.87 | 0.99 | 6 | - | - | - | - | - | - | - | - |
| Open-Source | One-Shot | 0.45 | 0.97 | 1.00 | 9 | - | - | - | - | - | - | - | - |
| Reasoning | One-Shot-CoT | 0.55 | 0.99 | 1.00 | 11 | - | - | - | - | - | - | - | - |
| **DeepSeek-V3** | Zero-Shot | 0.95 | 1.00 | 1.00 | 19 | - | - | - | - | - | - | - | - |
| 0324 (DeepSeek-AI et al., 2024) | Zero-Shot-CoT | 1.00 | 1.00 | 1.00 | 20 | - | - | - | - | - | - | - | - |
| Open-Source | One-Shot | - | - | - | - | - | - | - | - | - | - | - | - |
| Non-Reasoning | One-Shot-CoT | 0.90 | 1.00 | 1.00 | 18 | - | - | - | - | - | - | - | - |
| **DeepSeek-R1-671B** | Zero-Shot | 0.40 | 0.95 | 1.00 | 8 | 0.20 | 0.72 | 0.96 | 4 | - | - | - | - |
| 0120 (DeepSeek-AI, 2025) | Zero-Shot-CoT | 0.70 | 1.00 | 1.00 | 14 | - | - | - | - | - | - | - | - |
| Open-Source | One-Shot | 0.15 | 0.60 | 0.89 | 3 | 0.10 | 0.45 | 0.76 | 2 | - | - | - | - |
| Reasoning | One-Shot-CoT | 0.45 | 0.97 | 1.00 | 9 | 0.10 | 0.45 | 0.76 | 2 | - | - | - | - |
| **DeepSeek-R1-Distill-Llama-70B** | Zero-Shot | 0.10 | 0.45 | 0.76 | 2 | 0.05 | 0.25 | 0.50 | 1 | - | - | - | - |
| 0123 (DeepSeek-AI, 2025) | Zero-Shot-CoT | 0.40 | 0.95 | 1.00 | 8 | - | - | - | - | - | - | - | - |
| Open-Source | One-Shot | 0.70 | 1.00 | 1.00 | 14 | - | - | - | - | - | - | - | - |
| Reasoning | One-Shot-CoT | 0.35 | 0.92 | 1.00 | 7 | - | - | - | - | - | - | - | - |
| Total Summation | | | | | 540 | | | | 59 | | | | 46 |

Table 2: Evaluation of 10 LLMs and prompt techniques when generating formally verified code.

When investigating BRAK and SFLD, we observe in the results that the LLMs generate substantially fewer formally verified implementations than for SGMM. BRAK is solved 59 times in the 800 generated implementations. We suspect that the decrease in performance is attributed to the higher complexity of BRAK, as it contains nested conditional logic and input-validation requirements. Similarly, SFLD is not solved frequently. This module is solved 46 of 800 times, making SFLD the least solved module. The module contains timing constraints and indirect state-tracking requirements that complicate straightforward translation from specification to implementation. This suggests that, given our results, LLMs can reliably handle simpler modules but might struggle with more complex modules.

In our experiments, closed-source LLMs consistently generated more formally verified implementations than open-source LLMs across all modules. Interestingly, for BRAK, 59 of the 800 generated implementations verify: 49 by closed-source LLMs and only 10 by open-source LLMs (highlighted in column "$v_{s/20}$" for BRAK in Table 2). SFLD was solved exclusively by closed-source LLMs (46 total), with no open-source LLM generating a verified implementation. This pattern indicates that, within our study, closed-source LLMs hold a clear advantage in generating formally verified code.

Additionally, GPT-4.1, despite not being a reasoning LLM, performed well on all three modules in our experiments, showing that generating formally verified implementations does not depend solely on an explicitly reasoning-optimized architecture.

In summary, our results show that it is feasible to generate functionally correct implementations for all investigated safety-critical modules using LLMs. From conducted experiments, we observe that feasibility varies with module complexity: simpler modules, such as SGMM, are solved consistently, while success rates decline for more complex modules, including BRAK and SFLD. In our results, closed-source LLMs outperform open-source ones, but no single model solves all tasks. Interestingly, high performance is observed even for general-purpose LLMs without explicit reasoning optimization, suggesting that model capability combined with clear specifications is more critical than architectural specialization.

**RQ2 (prompting techniques influence)**: In Table 2, we observe that within the obtained results, the effectiveness of prompting techniques on generating formally verified implementations from specifications is highly LLM-dependent, with no universal "best" prompting technique. For example, `GPT-4.1` attains its highest SGMM success under One-Shot-CoT. Meanwhile, `Sonnet-3.7` performs best with Zero-Shot-CoT, and both `Qwen2.5-Coder-32B-Instruct` and `DeepSeek-R1-Distill-Qwen-32B` achieved the highest number of verified implementations in the experiments using Zero-Shot. For `DeepSeek-R1-Distill-Llama-70B`, the best-performing prompting technique for SGMM was One-Shot. The mentioned best-performing prompting techniques are highlighted in column "Prompting Technique" in Table 2 to illustrate the per-LLM differences. The mixed results suggest that the effectiveness of prompting strategies for generating formally verified code from specifications depends on the individual LLM.

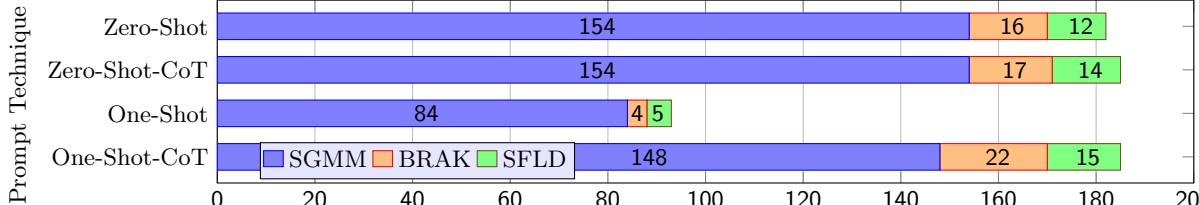

Figure 3: Count of formally verified implementations based on prompting techniques and modules.

Figure 3 presents a stacked bar plot illustrating the number of formally verified implementations generated across different prompting techniques and modules. The y-axis represents the four prompting techniques used. The x-axis indicates the total number of formally verified implementations. Each bar is divided into three segments, corresponding to the number of verified implementations for SGMM (blue), BRAK (orange), and SFLD (green). The values correspond to the aggregate counts provided in Table 2, with the total number of verified implementations per prompting technique being the sum of formally verified implementations across all LLMs.

Zero-Shot and Zero-Shot-CoT, and One-Shot-CoT all achieve very similar verification counts for the SGMM module, generating 154, 154, and 148 formally verified implementations, respectively. In contrast, One-Shot prompting produces substantially fewer SGMM verifications (84). For BRAK and SFLD modules, we see the same pattern where One-Shot performs significantly worse than other prompting techniques. For these two modules, One-Shot-CoT outperforms all other techniques in the experiments, generating 22 formally verified BRAK implementations and 15 SFLD implementations. The observations indicate that no single prompting technique uniformly excels across all LLMs. However, One-Shot prompting underperforms. We hypothesize that the lengthy example in a One-Shot prompt may overwhelm the input of the LLM, leading to poor performance. Additionally, we observe in Table 2 that for specific models, there is no significant difference between zero-shot and one-shot performance (o4-mini-high, both 20). In contrast, the performance of other models decreases significantly when using one-shot instead of zero-shot (Grok-3, from 20 to

1). But, when including the intermediate reasoning steps (CoT variants), this seems to clarify how to leverage the provided example in a more effective manner.

To conclude, the effectiveness of prompting varies markedly across LLMs and modules, and no single strategy consistently outperforms the others. Different LLMs may prefer a specific prompting technique for generating a formally verified implementation based on natural language and formal specifications. One-Shot alone performed substantially worse than other prompting techniques for all modules. However, the CoT variant resolves this decrease in performance for One-Shot.

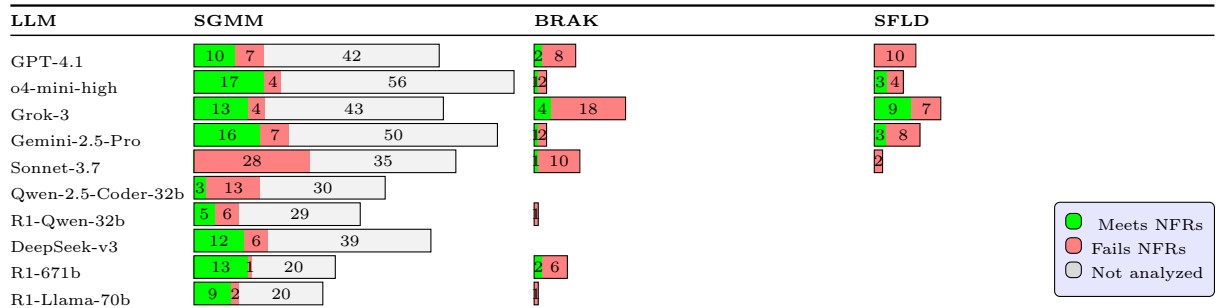

Table 3: Number of implementations adhering to all defined NFRs for each analyzed LLM. A total of 281 of 645 function implementations are analyzed. For brevity, the model names are shortened.

| Non-Functional Requirement | SGMM (of 176) | BRAK (of 59) | SFLD (of 46) |
| --- | --- | --- | --- |
| NFR4 (scope) | 1 | 41 | 22 |
| NFR7 (RTDB) | 63 | 26 | 10 |
| NFR9 (interface) | 20 | 1 | 16 |

Table 4: Number of violations of each NFR grouped by module. Non-violated NFRs are left out.

**RQ3 (non-functional compliance feasibility):**

For each LLM and module, we generate 80 implementations (4 prompting techniques × 20 implementations); only those that pass functional verification are checked for NFR compliance. The assessment of NFRs using the industrial static analysis tool revealed no violations, so those NFRs are omitted from further detailed analysis. Table 3 shows, per LLM and module, how many verified implementations also meet all NFRs. For the underrepresented modules (BRAK and SFLD), we analyze all implementations, whereas for SGMM, we evaluate only a random subset.

The results show that it is feasible to generate implementations that are functionally correct and adhere to NFRs for each of the evaluated modules. Within the analyzed samples, all LLMs, except for Sonnet-3.7, generate at least one implementation for SGMM that adheres to the NFRs. Interestingly, Sonnet-3.7 violates the NFR7 (RTDB) for all 28 formally verified implementations. For BRAK, six LLMs generate at least one implementation that adheres to both the requirement types. However, in the generated implementations, only o4-mini-high, Grok-3, and Gemini-2.5-Pro generate an implementation that is compliant with the NFRs for SFLDs. Additionally, these three LLMs generated at least one implementation that formally verifies and adheres to the defined NFRs, demonstrating feasibility.

NFR2 (loops) is never violated because the implementations for these modules do not involve loops. The other manually inspected NFRs are summarized in Table 4, which show the absolute number of violations for each of the manually inspected NFRs across the three modules. Column "Non-Functional Requirement" refers to the analyzed NFR as defined in Figure 2. Next, for each module, we present one column, presenting the number of observed NFR violations. The results show that NFR7 (RTDB) is violated most frequently, particularly for SGMM (63 of 176 times). Additionally, NFR4 (scope) is commonly violated in BRAK (41 of 59 times) and SFLD (22 of 46

times). When analyzing these implementations, we observe that the LLMs tend to group similar statements by defining constants at a broader scope. Adherence to NFR9 (interface) is less frequently violated, but still present, particularly in SGMM (20 of 176 times) and SFLD (16 of 46 times). Our results show that, in the conducted experiments, the LLM-generated implementations occasionally deviated from the defined NFRs. Overall, we observe that it is feasible that LLMs generate implementations that meet NFRs.

## 6. Discussion

As shown in Section 5, LLMs are capable of generating C module implementations that not only pass formal verification but also meet defined NFRs. Notably, the total cost of all conducted experiments across LLMs, prompting techniques, and modules amounted to \$ 321.87 (see Appendix E). This cost is substantially lower than the effort typically required from a human developer, who would likely spend many hours per module implementing, testing, and debugging the code. However, this promising performance comes with an important limitation: the engineering effort shifts from writing code to rigorously defining behavior through formal specifications and requirements, a process that is also costly and time-consuming.

In our experiments, we observed that when provided with specifications that clearly define the expected behavior for all input variables, LLMs are able to generate implementations that satisfy both functional and non-functional requirements. However, in typical development practice, engineers frequently work from incomplete or informal specifications and rely on their expertise to interpret and implement the intended behavior. Since LLMs lack this contextual understanding, the effectiveness of code generation depends heavily on the precision and completeness of the given specifications. Therefore, the long-term success of LLM-generated implementations from requirements relies not only on improving LLM capabilities but also on advancing methods and tools for writing rigorous and unambiguous specifications.

## 7. Conclusion

In this work, we evaluated ten state-of-the-art LLMs for generating formally verified, safety-critical automotive C programs across three industrial modules. Results demonstrate that it is feasible to generate programs that adhere to both functional and non-functional requirements. We observe that the performance of LLMs in generating functionally correct programs was not consistent throughout each analyzed module. Specifically, for the SGMM module, LLMs achieved a high success rate in generating a formally verified program (540 of 800). In contrast, verification rates were notably lower for modules containing either timing and state-tracking requirements, BRAK (59 of 800), and SFLD (46 of 800). While adherence to NFRs is feasible, our analysis revealed frequent violations, particularly in RTDB interaction (99 violations in 281 programs) and variable scope management (64 violations in 281 programs), highlighting areas for improvement in LLMs to generate programs that adhere to these requirements consistently. Future research should focus on developing automated methods for checking a broader range of NFRs. This would enable the construction of automatic feedback loops that can address violations and guide the LLM in iteratively generating improved programs. Furthermore, as the process of generating programs from specifications relies heavily on the quality of the requirements themselves, advancing tools and methodologies for authoring precise, complete, and unambiguous specifications remains a critical challenge. Finally, exploring fine-tuned LLMs on domain-specific software that adheres to functional and non-functional requirements could offer a promising direction for the automated generation of correct and standards-compliant automotive embedded code.

**Acknowledgments** This work has been partially funded by the Advanced Digitalisation Programme of Sweden's Innovation Agency (VINNOVA) as part of the FormAI project 2023-02671.

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

## Appendix A. Example of the inputs

This appendix gives an example of the inputs to and outputs from an LLM for a given module. To illustrate this, we present an implementation of a steering module, taken from Scania's development repository. This example module, named STEE, is a simplified version of a safety-critical automotive module that controls the steering system. It evaluates primary circuit sensors and determines whether the vehicle is in motion, and activates the secondary circuit under specific conditions. The input to the LLM is divided into five parts: (1) an ACSL specification, (2) natural language specifications, (3) a function signature, (4) type definitions, and (5) an interface definition. The desired output of the LLM is a single function implementation that adheres to the defined inputs.

### A.1. ACSL Specification

```
ACSL Specification

/*@
    assigns rtdb_PARKING_BRAKE_APPLIED;
    assigns rtdb_PRIMARY_CIRCUIT_HIGH_VOLTAGE;
    assigns rtdb_WHEEL_BASED_SPEED;
    assigns rtdb_SECONDARY_CIRCUIT_HANDLES_STEERING;
    assigns rtdb_ELECTRIC_MOTOR_ACTIVATED;

    assigns state_PARKING_BRAKE_APPLIED;
    assigns state_PRIMARY_CIRCUIT_LOW_FLOW;
    assigns state_PRIMARY_CIRCUIT_HIGH_VOLTAGE;
    assigns state_WHEEL_BASED_SPEED;
    assigns state_SECONDARY_CIRCUIT_HANDLES_STEERING;
    assigns state_ELECTRIC_MOTOR_ACTIVATED;

    // Req. 1 (combined conditions):
    // If old(rtdb_PRIMARY_CIRCUIT_HIGH_VOLTAGE) == 1
    //     OR old(rtdb_PRIMARY_CIRCUIT_LOW_FLOW) == 1,
    // AND old(rtdb_WHEEL_BASED_SPEED) > 3,
    // THEN rtdb_SECONDARY_CIRCUIT_HANDLES_STEERING == 1.
    ensures ((\old(rtdb_PRIMARY_CIRCUIT_HIGH_VOLTAGE) == 1)
            || (\old(rtdb_PRIMARY_CIRCUIT_LOW_FLOW) == 1))
            && (\old(rtdb_WHEEL_BASED_SPEED) > 3)
            ==> rtdb_SECONDARY_CIRCUIT_HANDLES_STEERING == 1;

    // Req. 2:
    // If rtdb_SECONDARY_CIRCUIT_HANDLES_STEERING == 1 (new state),
    // AND old(rtdb_PARKING_BRAKE_APPLIED) == 0,
    // THEN rtdb_ELECTRIC_MOTOR_ACTIVATED == 1.
    ensures rtdb_SECONDARY_CIRCUIT_HANDLES_STEERING == 1
            && \old(rtdb_PARKING_BRAKE_APPLIED) == 0
            ==> rtdb_ELECTRIC_MOTOR_ACTIVATED == 1;
*/
```

Figure 4: Formal ACSL Specification detailing the intended functionality of the STEE module.

Figure 4 presents the ACSL specification for the STEE module, which defines the expected behavior of the implementation using a precise format. It contains a function contract for the function for which we wish to generate an implementation. These formal specifications enable formal verification using Frama-C, ensuring that a given implementation complies with the intended requirements. The ACSL specification consists of three key components: requires, assigns, and ensures clauses.

The requires clause (not present in the example but commonly included in ACSL) defines preconditions that must hold before the function executes. The assigns clause explicitly lists the variables the function is allowed to modify. By restricting write operations, this clause ensures that the function does not write to variables other than the defined ones. The ensures clause defines postconditions that must be satisfied after function execution. These ensure clauses formally specify the conditions under which certain system states should change. In the formal specification of STEE, the ensures clauses correspond to the natural language requirements.

By providing a formal representation of system behavior, the ACSL specification enables automated verification using tools such as Frama-C. Additionally, this is given as input to the LLM to provide a formal means of defining the problem. The Frama-C plugins verify that the generated implementations meet the formal specification. The formal specification for STEE utilizes a total of 13 clauses to capture the behavior (11 assigns, 2 ensures).

## A.2. Natural Language Requirements

| Natural Language specification |
| --- |
| Req. 1: If the value of rtdb_PRIMARY_CIRCUIT_HIGH_VOLTAGE is 1 or the value of rtdb_PRIMARY_CIRCUIT_LOW_FLOW is 1), and the value of rtdb_WHEEL_BASED_SPEED is greater than 3, then rtdb_SECONDARY_CIRCUIT_HANDLES_STEERING must be set to 1 in the new state. |
| Req. 2: If rtdb_SECONDARY_CIRCUIT_HANDLES_STEERING is 1 in the new state, and the value of rtdb_PARKING_BRAKE_APPLIED is 0, then rtdb_ELECTRIC_MOTOR_ACTIVATED must be set to 1 in the new state. |

Figure 5: Natural language specifications detailing the intended functionality of the STEE module.

Figure 5 presents the natural language requirements of the STEE module. These requirements specify conditions for activating the secondary steering circuit and the electric motor. The first requirement ensures that the secondary circuit takes over steering when the primary circuit fails while the vehicle moves. The second requirement ensures the electric motor is engaged when the secondary circuit is active, and the parking brake is not applied.

## A.3. Function Signature

| Function Signature |
| --- |
| `void stee_10ms(void)` |

Figure 6: The function signature of the STEE module.

Figure 6 presents the function signature for the STEE module. The function signature, *stee_10ms*, has a void return type and takes no parameters. This is due to the module using variables defined in the interface. Adherence to the function signature is essential for the formal verification process. The LLM must generate an implementation that precisely matches this signature. During the verification process, the formal specification is prepended to the function implementation. The verification automatically fails if a LLM generates an implementation that does not adhere to the function signature. The function signature is used to prepend the formal specification, and if not found, then the formal specification is not prepended. Therefore, we decide to fail any implementation that does not adhere to the function signature.

## A.4. Header file

---

*Type Definitions*

```
enum SENSOR_STATE
{
    WORKING,
    NO_FLOW,
    SHORT_CIRCUIT
};

struct VEHICLE_INFO
{
    int wheelSpeed;
    int parkingBrake;
    int primLowFlow;
    int primHighVoltage;
    int secondCircHandlesStee;
    int electricMotorAct;
};

enum SIGNAL
{
    PARKING_BRAKE_APPLIED,
    PRIMARY_CIRCUIT_LOW_FLOW,
    PRIMARY_CIRCUIT_HIGH_VOLTAGE,
    WHEEL_BASED_SPEED,
    SECONDARY_CIRCUIT_HANDLES_STEERING,
    ELECTRIC_MOTOR_ACTIVATED,
    NUM_SIGNALS
};
```

---

Figure 7: Type definitions for the STEE module.

Figure 7 presents the type definitions provided in the header file for the STEE module. This header file is supplied to the LLM to ensure that the implementation uses the same data types and defined constants. In the figure, we have three data structures to be used in the implementation. These type definitions provide the required types to be used in the LLM-generated implementation.

## A.5. Interface Definition

Figure 8 presents the interface that a generated implementation must adhere to. This includes means of interacting with the RTDB. The interaction with the RTDB is abstracted away using global variables. The implementation reads from and writes to these variables. An implementation must only read from read-only variables and only write to write-only variables. The interface in the STEE module defines two sets of variables: *rtdb_* prefixed variables represent RTDB signals, while *state_* prefixed variables act as local states used during execution.

## A.6. Steering Module Code

Figure 9 presents an example implementation of the STEE module that adheres to the provided inputs, including the function signature, interface, ACSL specification, and natural language requirements. More importantly, this implementation meets the formal specification, meaning it satisfies all constraints defined in the ACSL specification when checked using Frama-C. The solution uses 52 lines of code to implement the behavior listed in the requirements.

---

*Interface*

```
/* Global variables representing RTDB signals (for illustration) */
int rtdb_PARKING_BRAKE_APPLIED;
int rtdb_PRIMARY_CIRCUIT_LOW_FLOW;
int rtdb_PRIMARY_CIRCUIT_HIGH_VOLTAGE;
int rtdb_WHEEL_BASED_SPEED;
int rtdb_SECONDARY_CIRCUIT_HANDLES_STEERING;
int rtdb_ELECTRIC_MOTOR_ACTIVATED;

/* Shadow states for reading/writing the signals */
int state_PARKING_BRAKE_APPLIED;
int state_PRIMARY_CIRCUIT_LOW_FLOW;
int state_PRIMARY_CIRCUIT_HIGH_VOLTAGE;
int state_WHEEL_BASED_SPEED;
int state_SECONDARY_CIRCUIT_HANDLES_STEERING;
int state_ELECTRIC_MOTOR_ACTIVATED;
```

---

Figure 8: Interface for reading and writing the vehicle state, simulating an RTDB mechanism.

## Appendix B.  Prompt Structure

### B.1.  Zero-shot and One-shot prompts

---

*Structured Prompt 1*

```
{
  "persona": "AI assistant for code generation & formal verification",
  "instructions": [
    "Make use of module-specific type definitions, imports, or header information",
    "Define the purpose and functionality of the module before implementing the code",
    "Adhere to formal language specifications",
    "Ensure the code interface is intuitive and well-documented",
    "Implement function signature correctly",
    "Follow predefined code generation rules"
  ],
  "rules": [
    {"id": 1, "rule": "Use structured control flow"},
    {"id": 2, "rule": "Avoid loops, prefer explicit sequences"},
    {"id": 3, "rule": "Limit memory allocation to init phase"},
    {"id": 4, "rule": "Minimize variable scope"},
    {"id": 5, "rule": "Validate inputs & check return values"},
    {"id": 6, "rule": "Limit pointer usage"},
    {"id": 7, "rule": "Follow single-read, single-write pattern"},
    {"id": 8, "rule": "Use only standard C constructs"},
    {"id": 9, "rule": "Adhere to provided interface"},
    {"id": 10, "rule": "Avoid external imports"},
    {"id": 11, "rule": "No ghost variables or predicates"}
  ],
  "example_module": {
    "name": "Vehicle Steering Control System",
    "example_input_module": "/* Safety-critical power steering logic */",
    "example_output_module": "void stee_10ms() {
        ...
    }"
  },
  "prompt": "Generate a formally verified C module implementation.",
  "inputs": {
    "formal_specification": "{{input_formal_specification}}",
    "natural_language_specification": "{{input_natural_language_specification}}",
    "interface": "{{input_interface}}",
    "type_definitions": "{{input_type_definitions}}",
    "function_signature": "{{input_signature}}"
  }
}
```

---

Figure 10: Simplified structured prompt for generating implementations. Blue highlights the one-shot example, only present in the one-shot version. Red highlights the problem-specific input.

---

*Function Implementation*

```c
void stee_10ms(void)
{
    enum SENSOR_STATE prim_sensor;

    /* Read current RTDB signals into local state. */
    state_WHEEL_BASED_SPEED = rtdb_WHEEL_BASED_SPEED;
    state_PARKING_BRAKE_APPLIED = rtdb_PARKING_BRAKE_APPLIED;
    state_PRIMARY_CIRCUIT_LOW_FLOW = rtdb_PRIMARY_CIRCUIT_LOW_FLOW;
    state_PRIMARY_CIRCUIT_HIGH_VOLTAGE = rtdb_PRIMARY_CIRCUIT_HIGH_VOLTAGE;
    state_SECONDARY_CIRCUIT_HANDLES_STEERING = rtdb_SECONDARY_CIRCUIT_HANDLES_STEERING;
    state_ELECTRIC_MOTOR_ACTIVATED = rtdb_ELECTRIC_MOTOR_ACTIVATED;

    /* Determine if the primary circuit is working, has no flow, or is shorted. */
    if (state_PRIMARY_CIRCUIT_HIGH_VOLTAGE == 1)
    {
        prim_sensor = SHORT_CIRCUIT;
    }
    else if (state_PRIMARY_CIRCUIT_LOW_FLOW == 1)
    {
        prim_sensor = NO_FLOW;
    }
    else
    {
        prim_sensor = WORKING;
    }

    /* Check whether the vehicle is moving. */
    char vehicleIsMoving = (state_WHEEL_BASED_SPEED > 3) ? 1 : 0;

    /* Check if vehicle is moving without primary power steering. */
    char vehicleIsMovingWithoutPrimaryPowerSteering =
        (vehicleIsMoving == 1 && (prim_sensor == NO_FLOW
        || prim_sensor == SHORT_CIRCUIT)) ? 1 : 0;

    /* If the vehicle is moving without primary power steering,
    let secondary circuit handle steering. */
    if (vehicleIsMovingWithoutPrimaryPowerSteering == 1)
    {
        state_SECONDARY_CIRCUIT_HANDLES_STEERING = 1;
    }

    /* If the secondary circuit is handling steering and parking brake is not set,
    activate the electric motor. */
    if (state_SECONDARY_CIRCUIT_HANDLES_STEERING == 1 &&
        state_PARKING_BRAKE_APPLIED == 0)
    {
        state_ELECTRIC_MOTOR_ACTIVATED = 1;
    }

    /* Write updated states back to RTDB signals. */
    rtdb_SECONDARY_CIRCUIT_HANDLES_STEERING = state_SECONDARY_CIRCUIT_HANDLES_STEERING;
    rtdb_ELECTRIC_MOTOR_ACTIVATED = state_ELECTRIC_MOTOR_ACTIVATED;
}
```

Figure 9: Example implementation of STEE.

Figure 10 outlines a simplified version of the structured prompt used for both zero-shot and one-shot program implementation generation. This prompt guides the LLM using five key components: (1) a predefined persona, (2) instructions for the LLM, (3) a set of NFRs (detailed in Appendix D),

(4) a hand-crafted example module for one-shot prompting, and (5) structured input fields for module specifications.

The persona in the prompt defines the role of the LLM, specifying that it acts as an assistant for generating formally verified C. This persona helps frame the task for the LLM, emphasizing that the generated implementation should follow the input specifications.

The instructions guide the LLM to follow a structured program implementation generation process. They direct the LLM to utilize the provided module definitions, explicitly define the purpose and functionality of the code before implementation, and ensure that the generated implementation aligns with the given input definitions. Additionally, the instructions emphasize the importance of producing well-documented code. These instructions help the LLM generate more consistent, interpretable, and verifiable implementations.

The NFRs specified in the prompt are taken from Appendix D. They establish constraints that guide the LLM in generating safety-critical code. This is done by giving guidelines and recommendations for high-quality, safety-critical code, such as avoiding loops and limiting memory allocation. Additionally, these NFRs help generate implementations that compile and verify. For compilation and verification purposes, the LLM must utilize the function signature.

Next, the prompt outline includes an example. This example (highlighted in blue) is present in the one-shot example but is left out of the zero-shot prompt. The example is another high-quality, manually created module. This example module refers to the STEE module, as detailed in Appendix A. This example provides a reference for the LLM, illustrating how the given specifications should be translated into code. By presenting a correctly structured and verifiable function, the one-shot prompt aims to guide the LLM toward generating formally verified implementations.

The next part of the prompt defines the core task by instructing the LLM to generate the implementation based on these instructions. Lastly, the prompt includes the structured inputs, highlighted in red in Figure 10. These inputs specify the input information that varies per module when prompting the LLM. For each modul,e there is a different set of inputs, while the remaining part of the prompt remains the same when invoking an LLM.

## B.2. Chain of Thought prompts

Figure 11 presents the CoT prompt, which follows the same structured input format as the non-CoT zero-shot and one-shot prompts but introduces reasoning steps before code generation. This CoT variant is used in both zero-shot and one-shot configurations, where the difference remains in whether the example STEE module is provided. The objective of adding a reasoning to the prompt is to encourage the LLM to systematically analyze the module requirements, apply coding constraints, and verify the correctness of the generated implementation.

Before generating an implementation, the LLM is guided through module analysis, where it considers the purpose, formal specifications, interface structure, and function signature. This is followed by a rule application step, in which the predefined NFRs are mentioned. The next step, code implementation, instructs the LLM to apply structured control flow, ensure input validation, and create an implementation that follows the expected format. Finally, the verification step prompts the LLM to review its implementation, checking for adherence to all formal requirements. Please note that this is a validation step, and formal verification is handled after the LLM generates an implementation. The formal verification step is automatically performed based on the response of the LLM. The structured input fields remain unchanged from the standard prompt, with the key difference being the addition of this systematic reasoning process.

**Structued CoT Prompt 1**

```json
{
  "persona": "AI assistant specializing in code generation and formal verification",
  "reasoning_framework": {
    "step1_module_analysis": {
      "description": "Analyze the module requirements",
      "considerations": [
        "What is the core purpose and functionality of this module?",
        "What formal specifications must be followed?",
        "How should the code interface be structured?",
        "What is the required function signature?"
      ]
    },
    "step2_rules_application": {
      "description": "Apply predefined coding rules",
      "rules": [
        {"id": 1, "rule": "Use structured control flow"},
        {"id": 2, "rule": "Avoid loops, prefer explicit sequences"},
        {"id": 3, "rule": "Limit memory allocation to init phase"},
        {"id": 4, "rule": "Minimize variable scope"},
        {"id": 5, "rule": "Validate inputs & check return values"},
        {"id": 6, "rule": "Limit pointer usage"},
        {"id": 7, "rule": "Follow single-read, single-write pattern"},
        {"id": 8, "rule": "Use only standard C constructs"},
        {"id": 9, "rule": "Adhere to provided interface"},
        {"id": 10, "rule": "Avoid external imports"},
        {"id": 11, "rule": "No ghost variables or predicates"}
      ]
    },
    "step3_code_generation": {
      "description": "Generate the function",
      "steps": [
        "Implement the function signature correctly",
        "Apply structured control flow and rule constraints",
        "Validate all inputs and outputs",
        "Document key decisions"
      ]
    },
    "step4_verification": {
      "description": "Verify the implementation",
      "checkpoints": [
        "Does it meet all formal specifications?",
        "Does it align with functional and interface requirements?",
        "Have all constraints been followed?"
      ]
    }
  },
  "one_shot_example": {
    "name": "Steering Control Module",
    "example_input_module": "/* Example module logic */",
    "example_output_module": "void stee_10ms() { ... }"
  },
  "instructions": "Proceed step by step, documenting reasoning at each stage.",
  "inputs": {
    "formal_specification": "{{input_formal_specification}}",
    "natural_language_specification": "{{input_natural_language_specification}}",
    "interface": "{{input_interface}}",
    "type_definitions": "{{input_type_definitions}}",
    "function_signature": "{{input_signature}}"
  }
}
```

Figure 11: Chain-of-thought prompt structure for generating formally verified C code. Blue highlights the one-shot example, which is left out in the zero-shot version of this prompt. Red highlights the parts that change depending on the modules in the analysis.

## Appendix C. Full LLM Experiments Table

| Model | Prompt Type | SGMM | | | | BRAK | | | | SFLD | | | |
|---|---|---|---|---|---|---|---|---|---|---|---|---|---|
| | | v@1 | v@5 | v@10 | $v_{s/20}$ | v@1 | v@5 | v@10 | $v_{s/20}$ | v@1 | v@5 | v@10 | $v_{s/20}$ |
| **GPT-4o** | Zero-Shot | 1.00 | 1.00 | 1.00 | 20 | 0.05 | 0.25 | 0.50 | 1 | - | - | - | - |
| 2024-08-06 (Hurst et al., 2024) | Zero-Shot-CoT | 0.70 | 1.00 | 1.00 | 14 | - | - | - | - | - | - | - | - |
| Closed-Source | One-Shot | 0.70 | 1.00 | 1.00 | 14 | - | - | - | - | - | - | - | - |
| Non-Reasoning | One-Shot-CoT | 0.90 | 1.00 | 1.00 | 18 | - | - | - | - | - | - | - | - |
| **GPT-4.1** | Zero-Shot | 0.75 | 1.00 | 1.00 | 15 | 0.10 | 0.45 | 0.76 | 2 | 0.05 | 0.25 | 0.50 | 1 |
| 2025-04-14 (OpenAI, 2025a) | Zero-Shot-CoT | 0.85 | 1.00 | 1.00 | 17 | 0.05 | 0.25 | 0.50 | 1 | 0.15 | 0.60 | 0.89 | 3 |
| Closed-Source | One-Shot | 0.45 | 0.97 | 1.00 | 9 | 0.05 | 0.25 | 0.50 | 1 | 0.05 | 0.25 | 0.50 | 1 |
| Non-Reasoning | One-Shot-CoT | 0.90 | 1.00 | 1.00 | 18 | 0.30 | 0.87 | 0.99 | 6 | 0.25 | 0.81 | 0.98 | 5 |
| **o1** | Zero-Shot | 0.65 | 1.00 | 1.00 | 13 | - | - | - | - | 0.05 | 0.25 | 0.50 | 1 |
| 2024-12-17 (Jaech et al., 2024) | Zero-Shot-CoT | 0.55 | 0.99 | 1.00 | 11 | 0.05 | 0.25 | 0.50 | 1 | - | - | - | - |
| Closed-Source | One-Shot | 0.15 | 0.60 | 0.89 | 3 | - | - | - | - | - | - | - | - |
| Reasoning | One-Shot-CoT | 0.80 | 1.00 | 1.00 | 16 | 0.05 | 0.25 | 0.50 | 1 | - | - | - | - |
| **o3** | Zero-Shot | 0.75 | 1.00 | 1.00 | 15 | - | - | - | - | - | - | - | - |
| 2025-04-16 (OpenAI, 2025b) | Zero-Shot-CoT | 0.95 | 1.00 | 1.00 | 19 | 0.15 | 0.60 | 0.89 | 3 | 0.05 | 0.25 | 0.50 | 1 |
| Closed-Source | One-Shot | 0.60 | 1.00 | 1.00 | 12 | - | - | - | - | - | - | - | - |
| Reasoning | One-Shot-CoT | 1.00 | 1.00 | 1.00 | 20 | 0.20 | 0.72 | 0.96 | 4 | 0.05 | 0.25 | 0.50 | 1 |
| **o3-mini** | Zero-Shot | 0.65 | 1.00 | 1.00 | 13 | 0.50 | 0.98 | 1.00 | 10 | - | - | - | - |
| 2025-01-31 (OpenAI, 2025) | Zero-Shot-CoT | 0.70 | 1.00 | 1.00 | 14 | 0.55 | 0.99 | 1.00 | 11 | - | - | - | - |
| Closed-Source | One-Shot | 0.85 | 1.00 | 1.00 | 17 | 0.05 | 0.25 | 0.50 | 1 | 0.10 | 0.45 | 0.76 | 2 |
| Reasoning | One-Shot-CoT | 0.85 | 1.00 | 1.00 | 17 | 0.75 | 1.00 | 1.00 | 15 | - | - | - | - |
| **o4-mini** | Zero-Shot | 0.95 | 1.00 | 1.00 | 19 | - | - | - | - | 0.05 | 0.25 | 0.50 | 1 |
| 2025-04-16 (OpenAI, 2025b) | Zero-Shot-CoT | 0.90 | 1.00 | 1.00 | 18 | - | - | - | - | 0.10 | 0.45 | 0.76 | 2 |
| Closed-Source | One-Shot | 0.90 | 1.00 | 1.00 | 18 | 0.05 | 0.25 | 0.50 | 1 | - | - | - | - |
| Reasoning | One-Shot-CoT | 0.95 | 1.00 | 1.00 | 19 | - | - | - | - | 0.10 | 0.45 | 0.76 | 2 |
| **o4-mini-high** | Zero-Shot | 1.00 | 1.00 | 1.00 | 20 | - | - | - | - | 0.10 | 0.45 | 0.76 | 2 |
| 2025-04-16 (OpenAI, 2025b) | Zero-Shot-CoT | 0.95 | 1.00 | 1.00 | 19 | 0.05 | 0.25 | 0.50 | 1 | 0.05 | 0.25 | 0.50 | 1 |
| Closed-Source | One-Shot | 1.00 | 1.00 | 1.00 | 20 | 0.05 | 0.25 | 0.50 | 1 | - | - | - | - |
| Reasoning | One-Shot-CoT | 0.90 | 1.00 | 1.00 | 18 | 0.05 | 0.25 | 0.50 | 1 | 0.20 | 0.72 | 0.96 | 4 |
| **Grok** | Zero-Shot | 0.40 | 0.95 | 1.00 | 8 | - | - | - | - | - | - | - | - |
| beta (xAI, 2024) | Zero-Shot-CoT | 0.30 | 0.87 | 0.99 | 6 | - | - | - | - | - | - | - | - |
| Closed-Source | One-Shot | 0.20 | 0.72 | 0.96 | 4 | - | - | - | - | - | - | - | - |
| Non-Reasoning | One-Shot-CoT | 0.30 | 0.87 | 0.99 | 6 | - | - | - | - | - | - | - | - |
| **Grok-3** | Zero-Shot | 1.00 | 1.00 | 1.00 | 20 | 0.20 | 0.72 | 0.96 | 4 | 0.40 | 0.95 | 1.00 | 8 |
| beta (xAI, 2025) | Zero-Shot-CoT | 1.00 | 1.00 | 1.00 | 20 | 0.40 | 0.95 | 1.00 | 8 | 0.25 | 0.81 | 0.98 | 5 |
| Closed-Source | One-Shot | 0.05 | 0.25 | 0.50 | 1 | - | - | - | - | - | - | - | - |
| Reasoning | One-Shot-CoT | 0.95 | 1.00 | 1.00 | 19 | 0.50 | 0.98 | 1.00 | 10 | 0.15 | 0.60 | 0.89 | 3 |
| **Gemini-2.0-Flash** | Zero-Shot | 0.90 | 1.00 | 1.00 | 18 | 0.05 | 0.25 | 0.50 | 1 | - | - | - | - |
| 001 (Google, 2025a) | Zero-Shot-CoT | 0.65 | 1.00 | 1.00 | 13 | - | - | - | - | - | - | - | - |
| Closed-Source | One-Shot | 0.60 | 1.00 | 1.00 | 12 | 0.10 | 0.45 | 0.76 | 2 | - | - | - | - |
| Non-Reasoning | One-Shot-CoT | 0.65 | 1.00 | 1.00 | 13 | - | - | - | - | - | - | - | - |
| **Gemini-2.0-Flash-Thinking** | Zero-Shot | 0.10 | 0.45 | 0.76 | 2 | 0.15 | 0.60 | 0.89 | 3 | - | - | - | - |
| 1219 (Google, 2025b) | Zero-Shot-CoT | 0.25 | 0.81 | 0.98 | 5 | - | - | - | - | - | - | - | - |
| Closed-Source | One-Shot | 0.35 | 0.92 | 1.00 | 7 | - | - | - | - | - | - | - | - |
| Reasoning | One-Shot-CoT | 0.30 | 0.87 | 0.99 | 6 | 0.05 | 0.25 | 0.50 | 1 | - | - | - | - |
| **Gemini-2.0-Pro** | Zero-Shot | 0.70 | 1.00 | 1.00 | 14 | 0.05 | 0.25 | 0.50 | 1 | - | - | - | - |
| 02-05 (DeepMind, 2025c) | Zero-Shot-CoT | 0.70 | 1.00 | 1.00 | 14 | - | - | - | - | - | - | - | - |
| Closed-Source | One-Shot | 0.65 | 1.00 | 1.00 | 13 | - | - | - | - | - | - | - | - |
| Reasoning | One-Shot-CoT | 0.90 | 1.00 | 1.00 | 18 | - | - | - | - | - | - | - | - |
| **Gemini-2.5-Flash** | Zero-Shot | 0.70 | 1.00 | 1.00 | 14 | 0.20 | 0.72 | 0.96 | 4 | 0.05 | 0.25 | 0.50 | 1 |
| 04-17 (DeepMind, 2025a) | Zero-Shot-CoT | 0.65 | 1.00 | 1.00 | 13 | 0.10 | 0.45 | 0.76 | 2 | 0.05 | 0.25 | 0.50 | 1 |
| Closed-Source | One-Shot | 0.55 | 0.99 | 1.00 | 11 | 0.05 | 0.25 | 0.50 | 1 | - | - | - | - |
| Non-Reasoning | One-Shot-CoT | 0.65 | 1.00 | 1.00 | 13 | - | - | - | - | 0.15 | 0.60 | 0.89 | 3 |
| **Gemini-2.5-Pro** | Zero-Shot | 0.85 | 1.00 | 1.00 | 17 | 0.10 | 0.45 | 0.76 | 2 | 0.05 | 0.25 | 0.50 | 1 |

| | | | | | | | | | | | | | |
|---|---|---|---|---|---|---|---|---|---|---|---|---|---|
| 05-06 (DeepMind, 2025b) | Zero-Shot-CoT | 1.00 | 1.00 | 1.00 | 20 | - | - | - | - | 0.15 | 0.60 | 0.89 | 3 |
| Closed-Source | One-Shot | 0.85 | 1.00 | 1.00 | 17 | - | - | - | - | 0.20 | 0.72 | 0.96 | 4 |
| Reasoning | One-Shot-CoT | 0.95 | 1.00 | 1.00 | 19 | 0.05 | 0.25 | 0.50 | 1 | 0.15 | 0.60 | 0.89 | 3 |
| **Sonnet-3.5** | Zero-Shot | 1.00 | 1.00 | 1.00 | 20 | - | - | - | - | - | - | - | - |
| 20241022 (Anthropic, 2024) | Zero-Shot-CoT | 1.00 | 1.00 | 1.00 | 20 | - | - | - | - | - | - | - | - |
| Closed-Source | One-Shot | 1.00 | 1.00 | 1.00 | 20 | - | - | - | - | - | - | - | - |
| Reasoning | One-Shot-CoT | 1.00 | 1.00 | 1.00 | 20 | - | - | - | - | - | - | - | - |
| **Sonnet-3.7** | Zero-Shot | 1.00 | 1.00 | 1.00 | 20 | 0.10 | 0.45 | 0.76 | 2 | - | - | - | - |
| 20250219 (Anthropic, 2025) | Zero-Shot-CoT | 1.00 | 1.00 | 1.00 | 20 | 0.35 | 0.92 | 1.00 | 7 | 0.10 | 0.45 | 0.76 | 2 |
| Closed-Source | One-Shot | 0.15 | 0.60 | 0.89 | 3 | - | - | - | - | - | - | - | - |
| Reasoning | One-Shot-CoT | 1.00 | 1.00 | 1.00 | 20 | 0.10 | 0.45 | 0.76 | 2 | - | - | - | - |
| **Meta-Llama-3.3-70B-Instruct** | Zero-Shot | 0.55 | 0.99 | 1.00 | 11 | - | - | - | - | - | - | - | - |
| 1206 (Dubey et al., 2024) | Zero-Shot-CoT | 0.65 | 1.00 | 1.00 | 13 | - | - | - | - | - | - | - | - |
| Open-Source | One-Shot | 0.15 | 0.60 | 0.89 | 3 | - | - | - | - | - | - | - | - |
| Reasoning | One-Shot-CoT | 0.25 | 0.81 | 0.98 | 5 | - | - | - | - | - | - | - | - |
| **Qwen2.5-Coder-32B-Instruct** | Zero-Shot | 0.95 | 1.00 | 1.00 | 19 | - | - | - | - | - | - | - | - |
| 2024-11-12 (Yang et al., 2024) | Zero-Shot-CoT | 0.50 | 0.98 | 1.00 | 10 | - | - | - | - | - | - | - | - |
| Open-Source | One-Shot | 0.40 | 0.95 | 1.00 | 8 | - | - | - | - | - | - | - | - |
| Reasoning | One-Shot-CoT | 0.45 | 0.97 | 1.00 | 9 | - | - | - | - | - | - | - | - |
| **DeepSeek-R1-Distill-Qwen-32B** | Zero-Shot | 0.70 | 1.00 | 1.00 | 14 | 0.05 | 0.25 | 0.50 | 1 | - | - | - | - |
| 0121 (DeepSeek-AI, 2025) | Zero-Shot-CoT | 0.30 | 0.87 | 0.99 | 6 | - | - | - | - | - | - | - | - |
| Open-Source | One-Shot | 0.45 | 0.97 | 1.00 | 9 | - | - | - | - | - | - | - | - |
| Reasoning | One-Shot-CoT | 0.55 | 0.99 | 1.00 | 11 | - | - | - | - | - | - | - | - |
| **DeepSeek-V3** | Zero-Shot | 0.95 | 1.00 | 1.00 | 19 | - | - | - | - | - | - | - | - |
| 0324 (DeepSeek-AI et al., 2024) | Zero-Shot-CoT | 1.00 | 1.00 | 1.00 | 20 | - | - | - | - | - | - | - | - |
| Open-Source | One-Shot | - | - | - | - | - | - | - | - | - | - | - | - |
| Non-Reasoning | One-Shot-CoT | 0.90 | 1.00 | 1.00 | 18 | - | - | - | - | - | - | - | - |
| **DeepSeek-R1-671B** | Zero-Shot | 0.40 | 0.95 | 1.00 | 8 | 0.20 | 0.72 | 0.96 | 4 | - | - | - | - |
| 0120 (DeepSeek-AI, 2025) | Zero-Shot-CoT | 0.70 | 1.00 | 1.00 | 14 | - | - | - | - | - | - | - | - |
| Open-Source | One-Shot | 0.15 | 0.60 | 0.89 | 3 | 0.10 | 0.45 | 0.76 | 2 | - | - | - | - |
| Reasoning | One-Shot-CoT | 0.45 | 0.97 | 1.00 | 9 | 0.10 | 0.45 | 0.76 | 2 | - | - | - | - |
| **DeepSeek-R1-Distill-Llama-70B** | Zero-Shot | 0.10 | 0.45 | 0.76 | 2 | 0.05 | 0.25 | 0.50 | 1 | - | - | - | - |
| 0123 (DeepSeek-AI, 2025) | Zero-Shot-CoT | 0.40 | 0.95 | 1.00 | 8 | - | - | - | - | - | - | - | - |
| Open-Source | One-Shot | 0.70 | 1.00 | 1.00 | 14 | - | - | - | - | - | - | - | - |
| Reasoning | One-Shot-CoT | 0.35 | 0.92 | 1.00 | 7 | - | - | - | - | - | - | - | - |
| Total Summation | | | | | 1172 | | | | 122 | | | | 61 |

Table 5: Evaluation of 22 LLMs and prompt techniques when generating formally verified implementations.

## Appendix D. Defined Rules

The NFRs used in this paper are derived from the Rules of Ten Holzmann (2018) and further refined based on observations of LLM-generated code. The Rules of Ten provide guidelines for writing safety-critical software, ensuring structured control flow, memory safety, and robustness. However, not all of the Rules of Ten apply to our setting. For example, we ask the LLM to generate entire function implementations for modules rather than employing subfunctions. Therefore, a subset of the Rules of Ten was selected for our selection of NFRs.

For each of the rules of the Rules of Ten, we selected whether this NFR is relevant to the function implementation task at hand. Four NFRs were removed from the Rules of Ten. Firstly, the NFR restricting implementation length was removed, as the focus is on generating complete modules rather than ensuring brevity in individual functions. Secondly, the requirement for a minimum assertion density was omitted, as our approach relies on formal verification rather than runtime assertions for correctness checking. Thirdly, constraints on preprocessor usage were deemed

unnecessary, as LLMs are not responsible for compiling code and do not need to manage complex macro definitions. Lastly, the NFR related to mandatory compiler warnings and static analysis checks was removed, as this is not useful for the LLM since we automatically do the compilation and verification. The remaining six NFRs from the Rules of Ten remain unchanged (NFRs 1–6).

Beyond adapting existing guidelines, additional NFRs are introduced based on observed failure patterns in LLM-generated code. Specific recurring issues, such as improper RTDB interactions, inconsistent adherence to the provided interface, and reliance on non-standard C constructs, were identified during experimentation. We aim to generate as many formally verifying implementations as possible; therefore, we introduce these NFRs to help the LLM generate implementations in the correct format.

The first added NFR, NFR7 (RTDB), is formulated to enforce structured RTDB access. We have abstracted the usage of RTDB to global variables. In practice, at the beginning of a function, one reads from the RTDB, then does the computation, and then writes to the RTDB. LLM-generated implementations often read from the RTDB anywhere in the code; we added this NFR to let the LLM use the RTDB correctly.

NFR8 (constructs) ensures that the LLM only uses standard C types or explicitly defined constructs. This NFR also closely corresponds with NFR10 (import). When generating implementations, the LLM frequently uses undefined types, such as bool, which is imported from an external source. The types defined in the input already contain a boolean definition, and we would like the LLM to use this type instead. Therefore, we tell the LLM to use the defined types (NFR8) and not use external imports (NFR10).

NFR9 (interface) mentions that the LLM uses the interface definitions. This NFR is added to ensure the LLM reads and writes to the correct variables. During manually performed experiments, the LLM did not consistently utilize these variables. NFR11 (formal specification logic) is added to improve the verification of the LLM-generated implementations. Ghost variables and predicates can only be used in the formal specification, not in the implementation code. As the LLM attempts to utilize these predicates and ghost variables, we explicitly mention that they cannot be used in the implementation of the LLM. These 11 NFRs guide the LLM in generating safety-critical code that we can formally verify.

## Appendix E. Cost Breakdown of LLM Usage

Table 6 reports the cost of generating implementations using each evaluated LLM. For every LLM, 240 implementations were generated (four prompting techniques × three modules × 20 implementations generated). The first column "Large Language Model" lists the names of the analyzed LLMs along with the specific version identifiers used in our experiments, typically denoting release dates or snapshot numbers to ensure reproducibility. Column "Reference" shows a reference, while the column "Cost" presents the total cost in United States dollar (USD) ($), calculated from actual input and output token usage as billed by the respective Application Programming Interfaces (APIs). Lastly, column "API Used" specifies the platform through which each LLM was accessed, which influences availability and pricing. Notably, some LLMs incurred no cost due to access via free-tier offerings. The overall cost of $321.87 USD for all experiments illustrates the affordability of large-scale LLM-based code generation relative to traditional software development.

| Large Language Model | Reference | Cost ($USD) | API Used |
|---|---|---:|---|
| GPT-4o (2024-08-06) | (Hurst et al., 2024) | 8.60 | OpenAI |
| GPT-4.1 (2025-04-14) | (OpenAI, 2025a) | 7.03 | OpenAI |
| o1 (2024-12-17) | (Jaech et al., 2024) | 124.29 | OpenAI |
| o3 (2025-04-16) | (OpenAI, 2025b) | 48.59 | OpenAI |
| o3-mini (2025-01-31) | (OpenAI, 2025) | 6.92 | OpenAI |
| o4-mini (2025-04-16) | (OpenAI, 2025b) | 6.27 | OpenAI |
| o4-mini-high (2025-04-16) | (OpenAI, 2025b) | 10.20 | OpenRouter |
| Grok (beta) | (xAI, 2024) | 22.25 | xAI |
| Grok-3 (beta) | (xAI, 2025) | 12.23 | xAI |
| Gemini-2.0-Flash (001) | (Google, 2025a) | 0.62 | Google DeepMind |
| Gemini-2.0-Flash-Thinking (1219) | (Google, 2025b) | 0 | Google DeepMind |
| Gemini-2.0-Pro (02-05) | (DeepMind, 2025c) | 0 | Google DeepMind |
| Gemini-2.5-Flash (04-17) | (DeepMind, 2025a) | 0.95 | Google DeepMind |
| Gemini-2.5-Pro (05-06) | (DeepMind, 2025b) | 14.85 | Google DeepMind |
| Sonnet-3.5 (20241022) | (Anthropic, 2024) | 17.15 | Anthropic |
| Sonnet-3.7 (20250219) | (Anthropic, 2025) | 28.96 | Anthropic |
| Meta-Llama-3.3-70b-Instruct (1206) | (Dubey et al., 2024) | 0.45 | OpenRouter |
| Qwen-2.5-Coder-32b-Instruct (2024-11-12) | (Yang et al., 2024) | 0.23 | OpenRouter |
| Deepseek-R1-Distill-Qwen-32B (0121) | (DeepSeek-AI, 2025) | 0.67 | OpenRouter |
| Deepseek-V3 (0324) | (DeepSeek-AI et al., 2024) | 0.37 | DeepSeek |
| Deepseek-R1-671b (0120) | (DeepSeek-AI, 2025) | 8.64 | OpenRouter |
| Deepseek-R1-Distill-Llama-70B (0123) | (DeepSeek-AI, 2025) | 2.60 | OpenRouter |
| Total | | 321.87 | |

Table 6: Cost breakdown of generating 240 function implementations per LLM, including version identifiers, total expenditure in USD, and APIs used.

