# OpenReview forum: "Generating Safety-Critical Automotive C-programs using LLMs with Formal Verification"
_nesyconf.org/NeSy/2025/Conference_Phase_2 — NeSy 2025 - Phase 2 Poster_

### Official Review · Reviewer_kbWE · 2025-06-23
**Valuable Insights into LLM-Based Code Generation, Conclusions Are Now More Supported and Less Overstated**

**Rating:** 7
**Confidence:** 4

**Review:**

The authors have broadly addressed the concerns raised in my initial review. The evaluation of non-functional requirements (RQ3) has been significantly expanded to include a more representative sample across model-module combinations, which improves the robustness of the corresponding conclusions. Additionally, the discussion and conclusion sections have been revised to adopt a more cautious tone, better reflecting the limitations observed in the data — particularly regarding the challenges posed by more complex modules. Overall, the revisions improve the clarity and credibility of the work, and I now find the paper suitable for acceptance.

**Anonymity:**

Remain anonymous

---

### Official Review · Reviewer_bV9S · 2025-07-02

**Rating:** 4
**Confidence:** 4

**Review:**

The authors evaluated LLM generated code in the context of automotive safety-critical software modules. The authors chose to evaluate the functional correctness of the generated code through functional verification, static analyzer and human evaluation. The authors prompted LLMs to generate code through 4 different prompting techniques for three modules. This research presents an interesting case study of dealing with LLM generated code in industrial application.

Pros:
1. The authors performed experiments with both proprietary and open weights models.
2. Data collected from human evaluation is novel and worth studying.
3. The revised discussion and conclusion section contains better discussions informing future research.

Cons:
1. With the advent of agentic coding, the 4 prompting techniques that the authors used to evaluate is now outdated. The validity and usefulness of these results are unclear to current practitioners who have moved on from these prompting techniques.
2. The authors separated CoT and non-CoT prompting techniques, but for reasoning models the CoT cannot be turned off. Perhaps the authors meant to contrast whether an example of the reasoning chain is provided within the prompt or not, but it is unclear in the paper.
3. The performance difference between Zero-Shot vs One-Shot remains under-explained. The authors merely "speculate" that the cause might be that One-Shot example poisons the context, but offers no evidence to back it up.
4. The speculated cause above also doesn't explain the mixed results of One-Shot-CoT vs Zero-Shot/Zero-Shot CoT where some modules perform better than others (because you can see from Table 1 that clearly some models perform the same for One-Shot vs Zero-Shot, whereas some models perform substantially worse, therefore it is more probable that it is a model-dependent issue rather than prompt technique-dependent issue).

Ultimately, the authors chose a research topic in which advancements are highly rapid, causing their results to be outdated in a short span of months. I appreciate that there wasn't enough time to redo the experiments, but including something like an iterative refinement prompting technique (which is similar to how agentic coding works) would have made the paper slightly more relevant to the current landscape.

**Anonymity:**

Remain anonymous

---

### Official Review · Reviewer_9D9m · 2025-07-09
**Generation of Safety-Critical Automotive C-programs using LLMs combined with Formal Verification**

**Rating:** 7
**Confidence:** 5

**Review:**

Summary:
    This paper is a new method for verified code generation, accounting for both functional and non-functional correctness. They create a system where an LLM accepts functional correctness specifications, natural language non-functional specifications, a function signature, type definitions, and an interface definition. The LLM generates an attempt to satisfy these conditions. This code is checked by compiler and then it is checked by a functional correctness verifier. The passing generations are collected and then non-functional specifications are partially confirmed by a static analysis tool, partially confirmed manually.
    Also, they performed experiments testing different prompt styles and different LLMs evaluating performance for the generation of code that passes each stage of verification.

Quality:
    The paper's method is sound and supported by empirical evidence from their experiments. They have some hypotheses about the reasons for degraded performance on different benchmarks which aren't fully verifiable/explored, but it is reasonable enough.

Clarity:
    The paper is clear in the presentation of their work. The content is well organized and easy to read and follow.

Significance:
    This work extends the existing approaches for verified LLM code generation by using static analysis tools to handle some non-functional correctness checking, along with manual verification for other non-functional properties. This area is moving quite fast and therefore there's some new state of the art changes to methodology not built upon, but the experiments and methodology are recent enough for the results to be applicable and useful.

Originality:
    The work is not exceptionally original, but is a decent extension of existing methods applied to LLM generation of verified code.

**Anonymity:**

Remain anonymous

---

### Official Review · Reviewer_aWCp · 2025-07-11
**Review - Formal Verification**

**Rating:** 7
**Confidence:** 3

**Review:**

### Strengths
- The paper demonstrated that in certain circumstances, generalized models that are not tuned for reasoning can be superior to those that are, even if the underlying task involves heavy reasoning. This is quite a surprising result.
- The recognition that prompt styles can make a significant difference in verification performance, and the associated evaluation presented in the paper is useful information for further developments. I would be curious to see this expanded with additional styles beyond the four provided.
### Weaknesses
- While Verify@k is certainly a better metric than Pass@k, there could be more value in granular non-binary repeated scoring. While the eventual goal with formal verification is to ensure that an implementation adheres to all defined constraints in the specification, it is often the case that certain classes of constraints are valued higher than others. To use automotive as an example, most would probably agree that ensuring the latency of electronic braking is within its defined specification is significantly more important than ensuring that the frequency of windshield wipers are within their correct timing interval. By taking this approach, we could would gain the ability to evaluate and identify generated implementations that adhere to "more important" constraints. By extending it to vector-valued scoring, we could better identify specific classes of constraints that our LLM is struggling to fulfill. This kind of scoring would give us guidance as to how we should modify the training dataset at various intervals for more efficient training.
- I think there should be a larger set of modules than just the three that were evaluated. This way, we could get a better sense of how well the method described will generalize.

### Overall Recommendation:
- I recommend that this submission be accepted. It is an improvement on previous related work, and provides interesting experimental results. If there are revisions made, I would recommend that more modules be used as part of the analysis, and perhaps compare the benefits and drawbacks of different evaluation mechanisms.

**Anonymity:**

Remain anonymous